# Differences in Cancer-Specific Mortality after Trimodal Therapy for T2N0M0 Bladder Cancer according to Histological Subtype

**DOI:** 10.3390/cancers14235766

**Published:** 2022-11-23

**Authors:** Francesco Barletta, Stefano Tappero, Andrea Panunzio, Reha-Baris Incesu, Cristina Cano Garcia, Mattia Luca Piccinelli, Zhe Tian, Giorgio Gandaglia, Marco Moschini, Carlo Terrone, Alessandro Antonelli, Derya Tilki, Felix K. H. Chun, Ottavio de Cobelli, Fred Saad, Shahrokh F. Shariat, Francesco Montorsi, Alberto Briganti, Pierre I. Karakiewicz

**Affiliations:** 1Cancer Prognostics and Health Outcomes Unit, Division of Urology, University of Montréal Health Center, Montréal, QC H2X 0A9, Canada; 2Unit of Urology/Division of Oncology, Gianfranco Soldera Prostate Cancer Lab, IRCCS San Raffaele Scientific Institute, Vita-Salute San Raffaele University, 20132 Milan, Italy; 3Department of Urology, IRCCS Policlinico San Martino, 16132 Genova, Italy; 4Department of Surgical and Diagnostic Integrated Sciences (DISC), University of Genova, 16132 Genova, Italy; 5Department of Urology, University of Verona, Azienda Ospedaliera Universitaria Integrata di Verona, 37126 Verona, Italy; 6Martini-Klinik Prostate Cancer Center, University Hospital Hamburg-Eppendorf, 20246 Hamburg, Germany; 7Department of Urology, University Hospital Frankfurt, Goethe University Frankfurt am Main, 60590 Frankfurt am Main, Germany; 8Department of Urology, IEO European Institute of Oncology, IRCCS, 20141 Milan, Italy; 9Department of Urology, University Hospital Hamburg-Eppendorf, 20251 Hamburg, Germany; 10Department of Urology, Koc University Hospital, 34010 Istanbul, Turkey; 11Department of Urology, Comprehensive Cancer Center, Medical University of Vienna, 1090 Vienna, Austria; 12Department of Urology, Weill Cornell Medical College, New York, NY 10021, USA; 13Department of Urology, University of Texas Southwestern, Dallas, TX 75390, USA; 14Hourani Center for Applied Scientific Research, Al-Ahliyya Amman University, Amman 19328, Jordan

**Keywords:** SEER program, bladder cancer, trimodal therapy, histological variant

## Abstract

**Simple Summary:**

Trimodal therapy represents an accepted treatment option for non-metastatic muscle-invasive bladder cancer, which is an alternative to radical cystectomy. Evidence regarding trimodal therapy efficacy has predominantly, or even exclusively, been applied to urothelial carcinoma of the urinary bladder patients. To address this void, we tested for differences in cancer-specific mortality in trimodal therapy-treated bladder cancer patients, according to histological subtype, namely urothelial carcinoma vs. neuroendocrine carcinoma vs. squamous cell carcinoma vs. adenocarcinoma.

**Abstract:**

We aimed at assessing the impact of non-urothelial variant histology (VH), relative to urothelial carcinoma of the urinary bladder (UCUB), on cancer-specific mortality (CSM) in T2N0M0 bladder cancer patients treated with trimodal therapy (TMT). TMT patients treated for T2N0M0 bladder cancer were identified within the Surveillance, Epidemiology, and End Results database (2000−2018). Patients who underwent TMT received trans-urethral resection of the bladder tumor, chemotherapy, and radiotherapy. CSM-FS rates were tested using Kaplan–Meier plots and multivariable Cox-regression (MCR) models according to histological subtype: UCUB vs. neuroendocrine carcinoma vs. squamous cell carcinoma vs. adenocarcinoma. A total of 3846 T2N0MO bladder cancer patients treated with TMT were identified. Of these, 3627 (94.3%) harbored UCUB, while 105 (2.7%), 85 (2.2%), and 29 (0.8%) harbored neuroendocrine carcinoma, squamous cell carcinoma, and adenocarcinoma, respectively. In Kaplan–Meier analyses, 3-yr CSM-FS rates were 57% for UCUB, 51% for neuroendocrine carcinoma, 35% for squamous cell carcinoma, and 60% for adenocarcinoma (*p*-value < 0.0001). In MCR models, only squamous cell carcinoma exhibited higher CSM than UCUB (HR 1.98, 95%CI 1.5–2.61, *p*-value < 0.001). Despite the small number of observations, squamous cell carcinoma distinguished itself from UCUB based on worse survival in T2N0M0 patients after TMT.

## 1. Introduction

Trimodal therapy (TMT) represents an accepted treatment strategy for non-metastatic muscle-invasive bladder cancer, which is an alternative to radical cystectomy (RC) [1]. It consists of maximal trans-urethral resection of the bladder tumor (TURBT), chemotherapy, and radiotherapy [2]. TMT might be considered for patients not eligible for RC, or for select, highly motivate patients interested in bladder-sparing regimens. To date, no randomized trials have compared the efficacy of TMT vs. RC. However, prospective studies and retrospective comparative analyses have shown encouraging oncological TMT results relative to RC [3,4,5]. In particular, large single-institution data has shown favorable oncological results for TMT-treated patients diagnosed with organ-confined disease [6]. However, these observations have predominantly, or even exclusively, been of UCUB patients. In consequence, cancer control outcomes are largely unknown for non-urothelial variant histology (VH) patients treated with TMT [7,8]. We addressed this knowledge gap in T2N0M0 bladder cancer patients treated with TMT who harbored histological subtypes other than UCUB and compared those individuals to their UCUB counterparts.

We hypothesized that no survival differences would be recorded according to bladder cancer histological subtypes. We addressed this objective using the Surveillance, Epidemiology, and End Results (SEER) database (2000–2018).

## 2. Materials and Methods

### 2.1. Study Population

Within the SEER database (2000–2018) [9], we focused on patients 18 years or older with histologically confirmed bladder cancer (International Classification of Disease for Oncology [ICD-O-3] site code C67.0–67.6 and C67.8–67.9). We only considered T2N0M0 patients treated with TMT. Patients who underwent TMT received TURBT, chemotherapy, and radiotherapy. The four histological subtypes of grades included were UCUB, squamous cell carcinoma, adenocarcinoma, and neuroendocrine carcinoma. Cancer-specific mortality (death from bladder cancer) was defined according to the SEER cause-specific death classification. Exclusion criteria consisted of unavailable information about grade and histology as well as all autopsy, death certificates, and missing follow-up data.

### 2.2. Statistical Analyses

Statistical analyses focused on cancer-specific mortality free-survival (CSM-FS) using Kaplan–Meier analyses and multivariable Cox-regression (MCR) models. Results were stratified according to histological subtypes: UCUB vs. neuroendocrine carcinoma vs. squamous cell carcinoma vs. adenocarcinoma. In MCR models, covariates consisted of age, grade (low vs. high), and sex. All statistical tests were two-sided with a level of significance set at *p*< 0.05. Analyses were performed using the R software environment for statistical computing and graphics (version 4.2.1; http://www.r-project.org/; accessed on 25 August 2022).

## 3. Results

### 3.1. Descriptive Characteristics of the Study Population

The study population consisted of 3846 T2N0M0 bladder cancer patients treated with TMT between 2000 and 2018. Patient median age (Table 1) was 77 years (70–83). Histological subtype distribution was as follows: UCUB 3627 (94.3%), neuroendocrine carcinoma 105 (2.7%), squamous cell carcinoma 85 (2.2%), and adenocarcinoma 29 (0.8%).

### 3.2. Kaplan–Meier Analyses Assessing CSM-Free Survival

Kaplan–Meier analyses depicted CSM-FS rates in T2N0M0 patients treated with TMT (Figure 1). Specifically, 3-yr CSM-FS rates were 57%, 51%, 35%, and 60% for UCUB, neuroendocrine carcinoma, squamous cell carcinoma, and adenocarcinoma, respectively (*p*-value < 0.0001).

### 3.3. Multivariable Cox Regression Models Predicting CSM

In MCR models, relative to UCUB, only squamous cell carcinoma emerged as an independent predictor of higher CSM (HR 1.98, 95% CI 1.5–2.61, *p*-value < 0.001) after adjusting for all covariates (Table 2).

## 4. Discussion

Few reports compared cancer control outcomes after TMT between UCUB and non-urothelial VH patients [10,11]. We addressed this void and hypothesized that no differences would distinguish VH from UCUB in TMT-treated T2N0M0 patients. Our analyses led to several noteworthy observations.

First, of all 3846 patients, a very marginal proportion of non-urothelial VH patients was recorded. Specifically, we identified 105 (2.7%), 85 (2.2%), and 29 (0.8%) individuals harboring neuroendocrine carcinoma, squamous cell carcinoma, and adenocarcinoma, respectively. These observations indicate that only very large-scale epidemiological databases might allow for comparisons between VH and UCUB after TMT. Moreover, it also explains the lack of studies addressing differences in survival according to different histological subtypes after TMT, based on institutional databases. Additionally, the rarity of non-urothelial VH patients treated with TMT also indicates the limited confidence that the urological community places on bladder-sparing strategies in the context of non-UCUB histological subtypes. Indeed, data validating the efficacy of TMT, predominantly, or even exclusively, stem from UCUB patients. In consequence, the use of TMT in non-UCUB patients cannot be based on strong objective evidence. These observations motivated the conduct of the current study.

Second, in Kaplan–Meier analyses, squamous cell carcinoma exhibited lower 3-yr CSM-FS than UCUB (35% vs. 57%). Moreover, in MCR models, squamous cell carcinoma represented an independent predictor of higher CSM (HR 1.98, 95%CI 1.5–2.61, *p*-value < 0.001), relative to UCUB. In consequence, our observations suggest that the use of TMT in T2N0M0 squamous cell carcinoma patients is associated with worse cancer control outcomes vs. their T2N0M0 UCUB counterparts. However, it should be emphasized that our findings are based on a relatively low number (n = 85) of squamous cell carcinoma patients. This limitation also applies to other non-UCUB histological subtypes. Specifically, only 105 neuroendocrine carcinoma and 29 adenocarcinoma patients were identified. Those histological subtypes did not reach independent predictor status for CSM, relative to UCUB. In consequence, our observations indicate worse cancer control outcomes in T2N0M0 squamous cell carcinoma patients treated with TMT, according to sufficient numbers of observations to allow valid statistical testing of potential CSM differences. However, comparisons of other histological subtypes to UCUB are based on marginal, if not insufficient, numbers of observations to justify valid conclusions.

We tested for cancer control differences in T2N0M0 patients treated with TMT, according to histological subtype, within a relatively large population-based cohort. Despite the large size of the current cohort, we encountered critical limitations due to sample size when non-UCUB histological subtypes other than squamous cell carcinoma were considered. Other analyses that addressed TMT in non-UCUB patients were affected by similar sample size limitations. For example, within an NCDB (2004–2013) analysis, Fischer-Valuck et al. [11] directly compared squamous cell carcinoma (n = 78) vs. UCUB (n = 3252) patients treated with TMT and detected worse median overall survival in squamous cell carcinoma patients (15.1 vs. 30.4 months; *p*-value = 0.013). Unfortunately, unlike in the current study, the authors did not include other non-UCUB histological subtypes to allow comparisons relative to UCUB. Therefore, our results regarding other histological subtypes cannot be directly compared to the Fischer-Valuck study [11]. However, the comparison of squamous cell carcinoma vs. UCUB patients after TMT revealed similar survival disadvantage, as recorded in the current study, even though a somewhat different endpoint was considered, which was the overall survival in NCDB vs. CSM in the current study. Unfortunately, NCDB only allows overall survival analyses and CSM cannot be distinguished from other-cause mortality. In a different study, which also focused on TMT-treated patients, Janopaul-Naylor et al. [10] (NCDB 2004–2015) tested for overall survival differences in RC- and TMT-treated patients according to different histological subtypes. The authors reported worse overall survival for squamous cell carcinoma TMT-treated patients (n = 94, HR 1.49, 95% CI 1.25–1.77, *p* < 0.001) as well as for adenocarcinoma TMT-treated patients (n = 38, HR 1.75, 95% CI 1.36–2.25, *p* < 0.001) vs. their RC counterparts harboring the same histological subtype. Again, the different study design, population, and endpoint do not allow direct comparisons with the current study. Furthermore, Krasnow et al. [12] compared outcomes of pure vs. variant UCUB patients after TMT. Specifically, 66 individuals (22%) harbored variant UCUB. Of these, 24 patients exhibited squamous differentiation (36%). In MCR models, variant UCUB was not associated with worse disease specific survival (*p* = 0.3). However, our findings are not comparable with Krasnow et al. [12] since the authors addressed differences in patients with UCUB according to variant histology differentiation. Moreover, patients exhibiting dominant histological subtypes different from UCUB (e.g., adenocarcinoma, squamous cell carcinoma) were excluded from the analyses. Unfortunately, to the best of our knowledge, no institutional or multi-institutional studies directly addressed the comparison made in the current study, namely CSM after TMT in T2N0M0 patients according to histological subtype.

Taken together, our study identified a very low number of T2N0M0 non-UCUB patients treated with TMT (n = 219). Of those, the majority harbored neuroendocrine carcinoma (n = 105), followed by squamous cell carcinoma (n = 85) and adenocarcinoma (n = 29) vs. 3627 UCUB patients. Squamous cell carcinoma patients distinguished themselves from UCUB patients based on significantly worse survival (3-yr CSM-FS 35% vs. 57%, MCR HR 1.98, *p*-value < 0.001). Previous reports showed marginal benefit from perioperative chemotherapy for squamous cell carcinoma patients treated with RC [13,14]. Thus, the radiosensitizing effect of chemotherapy might not be sufficient in squamous cell carcinoma TMT-treated patients to achieve adequate disease control. However, for other non-urothelial VH, the number of observations were insufficient to perform valid testing and interpretation of results. In consequence, it may be postulated that in TMT-treated T2N0M0 squamous cell carcinoma patients, significantly less favorable cancer control outcomes may be expected than when TMT is applied to their UCUB counterparts. In this context, novel therapy regimens (e.g., checkpoint inhibitors) and risk stratification based on genomic profiling might improve squamous cell carcinoma TMT treated patients’ cancer control outcomes [15].

Despite the novelty of our findings, several limitations need to be acknowledged. The first and foremost limitation consists of patient origin. Specifically, our findings are applicable to individuals who were captured within the SEER database. Therefore, the observations made within the current study cannot be applied to patients originating from outside of the United States or even patients that are not comparable to those included in the SEER database. For example, patients treated at centers of excellence, such as the Memorial Sloan Kettering Cancer Center or MD Anderson Cancer Center, are not included in the SEER database. Therefore, institutional or multi-institutional data reflecting cancer control outcomes of such individuals should be used if available. Second, our analyses relied on limited numbers of observations. Sample size represented a critical limitation, even within the current, very large-scale database. Consequently, it is unlikely that smaller scale databases, except for NCDB, will provide larger sample size results. Third, unlike other studies, we only focused on T2N0M0 bladder cancer patients. Such consideration was based on the concept that TMT is of marginal or no value in the presence of extra-vesical bladder cancer. Fourth, within the SEER database, only the dominant histological subtype is reported; therefore, analyses adjusting for presence of mixed histology variants cannot be assessed. Fifth, the SEER database does not provide specific information about size and focality of bladder tumors or completeness of TURBT. Similarly, the type, dose, and timing of radiotherapy as well as chemotherapy, were not available. Sixth, our endpoint consisted of CSM-FS. However, in TMT studies, endpoints may consist of bladder preservation, local recurrence with cystectomy or distant recurrence with preserved bladder. These endpoints cannot be addressed using SEER but are provided in institutional and multi-institutional analyses [6]. Finally, our report represents a retrospective analysis with high potential for selection biases.

## 5. Conclusions

Despite the small number of observations, squamous cell carcinoma distinguished itself from UCUB based on worse survival in T2N0M0 patients after TMT.

## Figures and Tables

**Figure 1 cancers-14-05766-f001:**
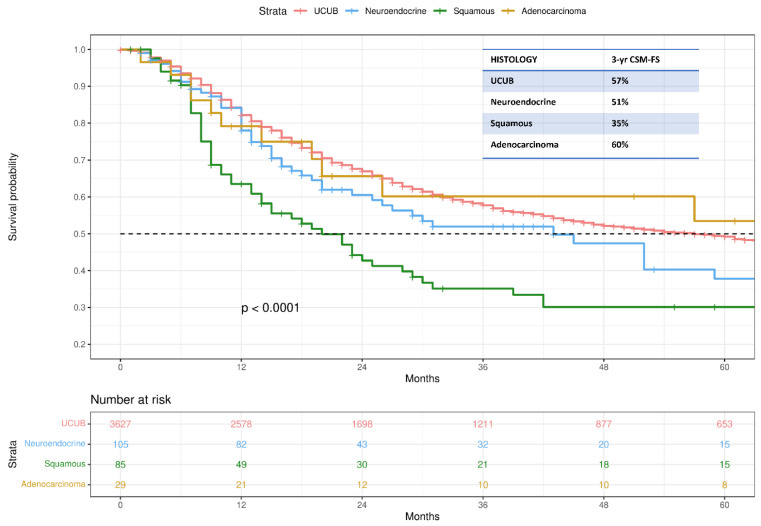
Kaplan–Meier analyses depicting cancer-specific mortality-free survival (CSM-FS) estimates of 3846 T2N0M0 bladder cancer patients identified within the Surveillance, Epidemiology, and End Results database (2000–2018). Data were stratified according to histology; UCUB: urothelial carcinoma of the urinary bladder.

**Table 1 cancers-14-05766-t001:** Descriptive characteristics of 3846 T2N0M0 bladder cancer patients treated with trimodal therapy identified within the Surveillance, Epidemiology, and End Results database (2000–2018).

Variables	Overall (n = 3846)
**Age at diagnosis (years)**	
Median	77
IQR	70–83
**Histology**	
Urothelial Carcinoma	3627 (94.3)
Neuroendocrine Carcinoma	105 (2.7)
Squamous Cell Carcinoma	85 (2.2)
Adenocarcinoma	29 (0.8)
**Grade**	
Low grade	113 (3)
High grade	3733 (97)
**Sex**	
Male	2842 (74)
Female	1004 (26)
**Tumor Site**	
Trigone	293 (7.7)
Dome	201 (5.2)
Lateral wall	871 (22.6)
Anterior wall	139 (3.6)
Posterior wall	328 (8.5)
Bladder neck	139 (3.6)
Ureteral orifice	76 (2)
Lateral-posterior wall	585 (15.2)
NOS	1214 (31.6)
**Follow-up (months)**	
Median	21
IQR	10–46

IQR: interquartile range; NOS: not otherwise specified.

**Table 2 cancers-14-05766-t002:** Multivariable Cox-regression models testing predictors of cancer-specific mortality.

Predictors	HR (95%CI)	*p*-Value
**Histology**		
Urothelial Carcinoma	Ref.	-
Neuroendocrine Carcinoma	1.28 (0.96–1.7)	0.09
Squamous Cell Carcinoma	1.98 (1.5–2.61)	<0.001
Adenocarcinoma	1.02 (0.58–1.8)	0.9
**Age at diagnosis (years)**	1.02 (1.02–1.03)	<0.001
**Sex**		
Female	Ref.	-
Male	0.88 (0.78–0.97)	0.02
**Grade**		
Low grade	Ref.	-
High grade	1.08 (0.82–1.42)	0.5

HR: hazard ratio; CI: confidence interval.

## Data Availability

All data generated for this analysis were from the Surveillance, Epidemiology, and End Results Research Plus (SEER) database. The code for the analyses will be made available upon request.

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
