# Peer review of "Differences in Cancer-Specific Mortality after Trimodal Therapy for T2N0M0 Bladder Cancer according to Histological Subtype"

_cancers, 2022, doi:10.3390/cancers14235766_

Round 1

Reviewer 1 Report

Congratulations on your paper. The paper is well written and results are clearly presented, but some improvements are required.  The paper has certain limitations which are elegantly emphasized in the discussion by the authors.

Authors should comment on the following issues and correct the manuscript accordingly:

1. Relatively low number of patients with histology variants was analyzed in the paper despite the population character of the study. This indicates that such dilemma to use TMT for such patients occurs very rarely.

2. Why did the authors include C67.7 ICD code? These are urachal tumours that are very rare and seem to have different histology and distinct prognosis. Tumour locations should also be listed in the table with baseline characteristics.

3. Which SEER variable was used to determine cancer-specific death? "SEER cause-specific death" or "COD to site recode". It sometimes differs and can be misleading.

4. It seems very unlikely that high-grade is not a CSM risk factor in MCR? Please comment on that. How was high-grade defined based on SEER variables?

5. Could you hypothesize why SCC has worse CSM after TMT? Why is the response to TMT worse? Please comment on that in the discussion.

6. In the discussion You did not mention an important study in that area by Krasnov et al.

“Clinical Outcomes of Patients with Histologic Variants of Urothelial Cancer Treated with Trimodality Bladder-sparing Therapy” Eur Urol. 2017 Jul;72(1):54-60. doi: 10.1016/j.eururo.2016.12.002.

That paper does not support the view of a worse prognosis in SCC after TMT and provides long-term survival analysis. Please mention this in the discussion.

Reviewer 2 Report

Barletta et al. present a very interesting topic of the prediction power of histology subtypes on the mortality of T2N0M0 bladder cancer patients after trimodal therapy. They did a comprehensive analysis of the patients with different histology subtypes. It turns out patients with squamous cell carcinoma displayed the worst cancer-specific mortality.

In general, this study is properly designed, and the results are clear and sound. However, it might be helpful if removing Table 1 as it is too simple, and the contents is clearly described in result section. It is also good to see survival analysis using data from other source, such as TCGA, etc. Some contents in Discussion section are simply describe the results, so can be move to Result section.

Round 2

Reviewer 1 Report

Thank You, I have no further comments